# Wave Function Realization of a Thermal Collision Model

**DOI:** 10.3390/e24121808

**Published:** 2022-12-12

**Authors:** Uriel Shafir, Ronnie Kosloff

**Affiliations:** The Institute of Chemistry, The Hebrew University of Jerusalem, Jerusalem 9190401, Israel

**Keywords:** open quantum systems, quantum, stochastic, central limit theorem, collision model, master equation, Markovianity

## Abstract

An efficient algorithm to simulate dynamics of open quantum system is presented. The method describes the dynamics by unraveling stochastic wave functions converging to a density operator description. The stochastic techniques are based on the quantum collision model. Modeling systems dynamics with wave functions and modeling the interaction with the environment with a collision sequence reduces the scale of the complexity significantly. The algorithm developed can be implemented on quantum computers. We introduce stochastic methods that exploit statistical characteristics of the model such as Markovianity, Brownian motion, and binary distribution. The central limit theorem is employed to study the convergence of distributions of stochastic dynamics of pure quantum states represented by wave vectors. By averaging a sample of functions in the distribution we prove and demonstrate the convergence of the dynamics to the mixed quantum state described by a density operator.

## 1. Introduction

In reality, every quantum system is open, while an isolated system is an exception. Therefore, the main setback in simulating and modeling a real-life quantum system is the high computational cost. To analyze the cost, we will first provide a generic description of an open quantum system, dive into the details of the dynamics, and finally observe the computational problem. After reviewing the current methods and their cost, we will demonstrate a scheme able to lower the computation complexity.

An open quantum system is generically described by the system’s Hamiltonian ${\hat{H}}_{s}$ and its density operator ${\hat{\rho }}_{s}$. The system is monitored by the measuring apparatus M. The environment is described by the Hamiltonian ${\hat{H}}_{B}$ and the interaction between the system and the environment is described by the Hamiltonian ${\hat{H}}_{SB}$. Contemporary open quantum systems include the IBM and Google quantum machines, constituting the nascent steps toward quantum computing [1]. If these devices are left alone, the quantum system ${\hat{\rho }}_{s}$ will reach thermal equilibrium with the extremely cold environment with a temperature $T_{B}$. The device is assembled from quantum circuits, which are constantly cooled. The computation output is measured by a measurement apparatus M. Another example of an open quantum system is the NV center in diamond [2,3]. These systems are constructed from a nitrogen impurity adjacent to a negative vacancy in a diamond. The primary quantum system Hamiltonian $H_{s}$ describes a spin triplet. The neighboring environment is composed of spins, such as other nitrogen impurities or carbon isotopes with a nuclear spin (${}^{13}C$). An additional environment is composed of lattice phonons. An optical measuring apparatus is coupled to the NV center able to measure changes in population, i.e., the measurement apparatus M [4].

Open system dynamics address a system interacting with the environment from the system’s perspective. Different approaches have been employed to construct reduced descriptions in terms of the system’s observable features. Starting with Bloch, a dynamical derivation based on the weak system bath coupling has led to the quantum master equation [5,6]. An alternative mathematical formulation employing quantum dynamical semigroups has yielded to a general structure termed the Gorini–Kossakowski–Lindblad–Sudarshan (GKLS) Equation [7,8]. Davis has linked the perturbation derivation to the general structure [9,10]. Non-Markovian formulations, including memory effects, have also been suggested [11].

Our mission is to develop a simulation algorithm for the dynamics of an open quantum system. In this approach, a system is viewed from a thermodynamic perspective. Recent studies by Dann et al. [12] have provided the conditions for obtaining consistency between Markovian dynamics and thermodynamic principles in open quantum systems. The consistency conditions are formulated by a set of axioms which can be applied to the present study.

Our desired quantum simulation allows implementation on both classical and quantum processors. Quantum processors are designed on a wave function’s formulation. In classical computers, the wave function description is computationally preferable to a density operator formalism.

The scale of computational resources for simulating quantum dynamics is at least polynomial with the size of the Hilbert space. In addition, every degree of freedom in the system increases the size of the Hilbert space exponentially. For example, in a system composed of spin particles, the size of the Hilbert space scales as $2^{n}$, where n is the number of particles. Describing the system with a density matrix squares the memory resources of the computation.

The computational resources for solving the dynamics and counting the number of operations scales with the product of the Hilbert space size multiplied by the product of propagation time and energy range [13]. The energy range is also doubled in the density operator description [14]. Another advantage in a wavefunction formulation is a pedagogical one. Describing the dynamics of a single pure state provides an intuitive sense of the process termed the quantum trajectory [15]. For these reasons, we have developed a simulation method describing the dynamics of open quantum systems in a wavefunction formulation.

The starting point for establishing the dynamics of an open quantum system is a global approach incorporating the system and the environment. Typically, one constructs a global Hamiltonian-$\hat{H}$, composed of the system (S), bath (B), and interaction (HB) Hamiltonians:(1)$$\hat{H}={\hat{H}}_{S}+{\hat{H}}_{B}+{\hat{H}}_{SB}.$$
Solving the Schrödinger equation for the combined system is computationally prohibited due to the enormous number of degrees of freedom. To overcome this obstacle, the entire setup is partitioned between the system and the environment. It is customary to diagonalize the bath Hamiltonian to orthogonal modes either harmonic or composed of an ensemble of spins [11,16]. The next step is obtaining effective reduced equations of motion for the system in which the bath enters implicitly. If the system–bath interaction induces entanglement necessarily the reduced description of the system will be described by a mixed state ${\hat{\rho }}_{s}$ [17].

Assuming an initial uncorrelated system and bath $\hat{\rho }=\rho_{S}⊗{\hat{\rho }}_{B}$, the system propagation becomes a completely positive trace-preserving dynamical map (CPTP) ${\hat{\rho }}_{S}\left(t\right)=\Lambda_{t}{\hat{\rho }}_{S}\left(0\right)$ [18]. Additionally, imposing the condition of Markovianity, Gorini, Kossakowski, Sudarshan and Lindblad derived the general form of the master equation [7,8]. The GKLS has become one of the cornerstones of the theory of open quantum systems.

Solving the master equation is a difficult computational problem. The state is described by density matrices resulting in a computational scaling of at least $O(N^{2})$, where *N* is the size of the Hilbert space.

To reduce the computational complexity, a wavefunction method is desirable. The algorithm involves a stochastic unraveling of wavefunctions. Stochastic approaches are currently in use in many fields of quantum dynamics, such as thermal averaging [19,20] and electronic structure methods [21]. A stochastic approach has been suggested by Percival and Gisin [22,23,24,25,26,27] for unraveling the GKLS equation. In their approach, the GKLS equation was transformed into stochastic differential equations. This unraveling procedure is non-unique. It also has the benefit of the freedom to use the most mathematically convenient choice. A drawback is that the stochastic wavefunction is not associated with a physical description. An additional problem of this method is that the dynamics are formulated by a non-linear differential equation. This enhances the difficulty of finding a solution approaching a computational scaling of the Hilbert space size squared. A different unraveling approach was developed by Katz, Torrontegui and Kosloff [28,29]. The method partitions the environment to primary and secondary baths. The primary bath, termed the surrogate Hamiltonian, is weakly coupled to the system [30,31]. Stochastically, the spins of the primary bath are refreshed from the secondary bath.

We now advance toward complexity reduction and focus on a method for unraveling wavefunctions using stochastic variables modeled on the collision model (CM). The quantum CM first appeared in 1948 in a paper by Karplus and Schwinger [32], followed by the work of J. Rau [33]. In the 1960s, and later on in the 1980s, CMs appeared in studies on weak measurements by C. M. Caves and G. J. Milburn [34,35]. In recent years, CMs have become more popular due to their simplicity, consistency with thermodynamics [36,37,38,39,40,41,42], and low computational effort in the bath description [43].

The collision model decomposes the bath to an ensemble of ancilla sub-units with which the system interacts. Consequently, there is one basic assumption in the general CM:The system interaction with the bath is described as an interaction between the system with and a single ancilla from the environment.

We develop a simple collision model with two additional assumptions regarding the bath:1.Ancillas do not interact with each other.2.Ancillas are initially uncorrelated.

The collision model can directly simulate real physical scenarios, such as collision with a diluted gas as well as an NV center interacting with a spin bath [29]. In addition, one can imagine a virtual collision. For example, an interaction with an electron–hole pair.

This general description of CM requires addressing two major complexity problems:1.In order to describe a system’s interaction with the ancilla, we must solve the interaction dynamics according to some physical model. This will require solving the time-dependent Schrödinger or Liouville equation, which is generally a very hard task that scales unfavorably for large systems.2.The density matrix size of a multi-particle system grows exponentially with the number of particles.

A significant simplification of the model is achieved by adding the assumption that the time period of interaction between the system and the bath ancilla is much faster than the typical free dynamics timescale of the system. This enables splitting of the propagation into the internal system propagation and dissipation emerging from the interaction with the bath.

The complexity is further reduced by an implicit treatment of the bath. We choose to describe the primary system as a system of coupled spins.

This representation is employed for three main reasons:1.It is computationally inexpensive.2.A system composed of qubits is universal and can therefore simulate other physical systems [44].3.Such a system can be implemented on a quantum computer.

The collision model generates a completely positive trace-preserving map. This process leads to a fixed point of the map which is associated with an equilibrium state in a physical system. In the CM, an implicit treatment of the bath is obtained by tracing the Hilbert space of the ancilla after interacting with the system. Thus, a crucial tool for simulation is a partial trace algorithm implemented in a wave functions formalism.

We have developed a stochastic algorithm implemented in a wave function formulation. The setup is based on the partial trace implementation. In addition, we obtain an intuitive description of the process as an average of partial measurements of the system. The methods presented in this paper can be implemented in any case of a collision model where the environment is composed of qubits, as in [45,46]. We stress that any open system represented by a collision model can be described by the method presented in this paper including the studies [40,47,48].

The stochastic wave function algorithm developed here is especially useful for multi-particle systems, which require significant resources to compute [49].

## 2. Implementation

### 2.1. Setup Description

The outline of the derivation assumes a unitary evolution generated by the total Hamiltonian:(2)$$\hat{H}={\hat{H}}_{S}+{\hat{H}}_{B}+{\hat{H}}_{SB},$$
composed of the system Hamiltonian ${\hat{H}}_{S}$, environmental Hamiltonian ${\hat{H}}_{B}$, and interaction ${\hat{H}}_{SB}$. We assume ℏ=1 throughout the paper.

### 2.2. Representation of the Spinor

The studied model is composed of a system of qubits. For such a system, the wave function has dimensions of $1\times 2^{N}$, where *N* is the number of qubits. The density matrix representing the system has the dimensions of $2^{N}\times 2^{N}$. To expand the wave function, we choose a local expansion constructed by the basis of individual components. The natural choice for qubits is to construct the wave function in a computational basis as a linear tensor product of the computation base spanning each qubit space [50]. Each qubit is represented in the computational basis, where $|0\rangle =|↓\rangle$ and $|1\rangle =|↑\rangle$. The complete basis on which we will represent the system is:(3)$$\left\{\Pi_{i=1}^{N}⊗|\delta_{i}\rangle \right\}$$
where $\delta_{i}=0,1$
(4)$$\left\{\Pi_{i=1}^{N}⊗|\delta_{i}\rangle \right\}=\left\{|00\ldots 0\rangle ,|00\ldots 1\rangle ,\ldots ,|01\ldots 0\rangle ,|01\ldots 1\rangle ,|10\ldots 0\rangle ,|10\ldots 1\rangle ,|11\ldots 0\rangle ,|11\ldots 1\rangle \right\}$$

The basis is ordered in a raising order of the binary basis.

### 2.3. System and Baths

The multi-qubit system Hamiltonian $\hat{H_{S}}$ is given as:(5)$$\hat{H_{S}}=\sum_{k}\hat{H_{k}}+\sum_{i,j}\epsilon_{i,j}\left({\hat{\sigma }}_{{+}_{i,j}}+{\hat{\sigma }}_{-i,j}\right),\;$$
where $\hat{H_{k}}$ is the free *k*’th particle Hamiltonian, $\epsilon_{i,j}$ is the interaction coefficient between the *i* and *j* qubits, i.e, the interaction between the *i* and *j* particles in the system.

Where *N* is the number of particles and
$$\begin{array}{c} {\hat{\sigma }}_{+i,j}={\hat{I}}_{2^{i-1}}⊗|0\rangle \langle 1|⊗{\hat{I}}_{2^{j-i-1}}⊗|1\rangle \langle 0|⊗{\hat{I}}_{2^{N-j}}\\ {\hat{\sigma }}_{-i,j}={\hat{I}}_{2^{i-1}}⊗|1\rangle \langle 0|⊗{\hat{I}}_{2^{j-i-1}}⊗|0\rangle \langle 1|⊗{\hat{I}}_{2^{N-j}} \end{array}$$
${\hat{\sigma }}_{\pm i,j}$ are operators connecting the *i*’th and *j*’th particles. We represent the state of the system as the density operator ${\hat{\rho }}_{S}$.

The density matrix of the bath $\rho_{B}$ is composed of uncorrelated ancilla qubits: $\rho_{b}$.
(6)$${\hat{\rho }}_{B}={\hat{\rho }}_{b_{1}}⊗{\hat{\rho }}_{b_{2}}⊗\ldots ⊗{\hat{\rho }}_{b_{n}}=\Pi_{i}⊗{\hat{\rho }}_{b_{i}}=\Pi_{i}⊗\frac{e^{-\beta {\hat{H}}_{b_{i}}}}{Z}$$
where ${\hat{H}}_{b}$ is the Hamiltonian of the individual ancilla qubit and β is the inverse temperature times the Boltzman factor $\beta =\frac{1}{k_{B}T}$. The fact that the bath is uncorrelated with the system initially is in consent with the second postulate of Dann and Kosloff [12].

#### Observables

An observable 〈O〉 is defined as $tr\{\hat{O}\hat{\rho }\}$ in the density matrix formalism. For a pure state described by a wavefunction, it is also defined as $\langle \psi |\hat{O}|\psi \rangle$. Therefore, in a pure state, the two definitions are equivalent. Let $\left\{|\psi_{i}\rangle \right\}$ be an orthonormal basis with $|\psi_{k}\rangle =|\psi \rangle$ then
(7)$$tr\left\{\hat{O}\hat{\rho }\right\}=\sum_{i}^{n}\langle \psi_{i}|\hat{O}|\psi \rangle \langle \psi ||\psi_{i}\rangle =\langle \psi |\hat{O}|\psi \rangle$$

### 2.4. 
System Ancilla Interaction

The interaction between the system and the environment is represented as a repeated interaction between the system and a subsystem of the environment, typically a thermal qubit. A general unitary interaction is employed. Therefore, the interaction can be expressed by its generator, the interaction Hamiltonian:(8)$${\hat{U}}_{int}=e^{-i{\hat{H}}_{int}\theta }$$
where the phase angle θ has units of time. Because we can express an exponent as a polynomial sum of powers of $H_{int}$, we obtain
(9)$$[{\hat{U}}_{int},{\hat{H}}_{int}]=0.$$

### 2.5. Dynamics

In the collision model, we assume that the uncorrelated thermal qubit is employed only once. After an interaction, we trace out the Hilbert space of the ancilla qubit $\rho_{b}$. It is thus assumed that the bath’s state is unchanged. This assumption is with accordance with an uncorrelated infinitely large bath and imposes that the system’s state depends entirely on its previous state. The last remark is a definition of Markovianity and is in accordance with postulate 4 of Dan and Kosloff [12]
(10)$$\Lambda_{t}=\Lambda_{t-s}\Lambda_{s}$$
The dynamical map Λ propagates density operators. A reduced map, generated by a global Hamiltonian in Equation (Equation 2), from an initial uncorrelated state defines a Kraus map [18]. Such a map Λ is a completely positive trace-preserving (CPTP) map on the system [12]. The generator of the dynamics is defined as:(11)$$L=\lim_{t\to 0}\frac{\Lambda \left(t\right)-\hat{I}}{dt}$$
Under the assumption that the collision period is much shorter than the interval between collisions, we can write the generator as follows:(12)$$L=-i\left[\hat{H},•\right]+\gamma \left(tr_{b}\left\{{\hat{U}}_{int}•⊗{\hat{\rho }}_{b}{\hat{U}}_{int}^{†}\right\}-\hat{I}•\right)\;\;\;,$$
where γ is the collision rate, Equation (Equation 12). The reduced description will have the form
(13)$$\frac{d}{dt}{\hat{\rho }}_{S}=L\left({\hat{\rho }}_{S}\right)=-i\left[{\hat{H}}_{S},{\hat{\rho }}_{S}\right]+\gamma \left(tr_{b}\left\{{\hat{U}}_{int}{\hat{\rho }}_{S}⊗{\hat{\rho }}_{b}{\hat{U}}_{int}^{†}\right\}-{\hat{\rho }}_{S}\right)$$

This structure is known as the Poissonian GKLS form [7]. Assuming the interaction is instantaneous, we adopt a discrete form of the propagator where a repeated sequence of instantaneous collision following a free propagation, of the system is implemented. This integrated form of the dynamics breaks up the evolution to a sequence of collision events, where γ is the collision rate per unit time. A single collision event can be described by the super-operator M acting on the density operator ${\hat{\rho }}_{s}$.
(14)$$M\left({\hat{H}}_{S},\gamma ,\theta ,\beta \right){\hat{\rho }}_{S}=\left(\hat{U}\left({\hat{H}}_{S},dt\right)\right)tr_{b}\left\{{\hat{U}}_{int}\left(\theta \right){\hat{\rho }}_{S}⊗{\hat{\rho }}_{b}{\hat{U}}_{int}{\left(\theta \right)}^{†}\right)\}\left({\hat{U}}^{†}\left({\hat{H}}_{S},dt\right)\right)$$
Equation (Equation 14) describes a scheme of operation. An interaction ${\hat{U}}_{int}$ is acting on the combined uncorrelated system and ancilla particle ${\hat{\rho }}_{S}⊗{\hat{\rho }}_{b}$. Following this, a reduced description of the system alone is generated by the partial trace operation and as a last step, a free propagation $\hat{U}\left({\hat{H}}_{S},dt\right))$. When *k*-consecutive collisions are implemented, it results in the production of k super operators
(15)$$\Lambda_{s}=\prod_{i=1}^{k}M_{i}\left({\hat{H}}_{S},\gamma ,\theta ,\beta \right)\;\;\;.$$

### 2.6. Unraveling of the Density Operator

The density operator $\hat{\rho }$ completely describes the state of a quantum system. Any observable is determined by the relationship: $\langle O\rangle =tr\left\{\hat{\rho }O\right\}$. The density operator was introduced by von Neuman to describe statistical phenomena in quantum mechanics [17]. Von Neuman observed that a pure state in an entangled composite system is reduced to a mixed state when a local observation of a subsystem is performed. This statistical characteristic of the mixed state cannot be presented by a wave function formalization. The thermal state, by construction, is a statistical mixture and is thus also not describable by a single wavefunction. To overcome this issue, the open quantum system state and its dynamics are obtained by unraveling the density operator employing many quantum wavefunctions.

The statistical character of the density operator is reflected by unraveling to an average of an outer product of wave functions:(16)$$\hat{\rho }=\sum_{k}p_{k}|\psi_{k}\rangle \langle \psi_{k}|$$
where $\left\{|\psi_{k}\rangle \right\}$ is the set of unraveling wavefunctions (not necessarily orthogonal).

The unraveling set {|ψ〉} is not unique, allowing freedom that we will exploit. A straightforward unraveling is obtained by diagonalizing the density matrix $\hat{\rho }$: (17)$$\hat{\rho }=\sum_{n=1}^{N}p_{i}|n\rangle \langle n|$$ where $|n\rangle$ are the orthogonal eigenfunctions and $p_{i}$ are positive coefficients with $\sum_{i}p_{i}=1$ and thus can be interpreted as a probability set of a complete measurement of $\hat{O}$ which commutes with $\hat{\rho }$.

### 2.7. Stochastic Unraveling

The present study employs a stochastic unraveling scheme based on the flowing lemma: let θ be a random phase wavefunction then
(18)$$\lim_{K\to \infty }\frac{1}{K}\sum_{j,k=1}^{K}e^{i(\theta_{j}-\theta_{k})}=\delta_{j,k}$$
where *K* is the number of random phases.

Let $|\psi_{\theta }\rangle$ be a wave function composed of an equal superposition of an arbitrary orthonormal basis $\left\{|n\rangle \right\}$ of size N with random phase θ
(19)$$|\psi_{\theta }\rangle =\frac{1}{\sqrt{N}}\sum_{n=0}^{N-1}e^{i\theta_{n}}|n\rangle$$
From Lemma (Equation 18) the identity operator ${\hat{I}}_{N}$ can be resolved by an infinite sum of random wave functions.
(20)$$\lim_{K\to \infty }\frac{N}{K}\sum_{i=1}^{K}|\psi_{\theta_{i}}\rangle \langle \psi_{\theta_{i}}|={\hat{I}}_{N}$$

This allows unraveling of the density operator with stochastic wave functions. We will prove that for the basis $\left\{|n\rangle \right\}$ that diagonalizes $\hat{\rho }$, we can use Equation (Equation 19) to create a set of stochastic wave functions that unravel $\hat{\rho }$.
(21)$$|\phi \rangle =\sqrt{N}\sum_{n=1}^{N}\sqrt{p_{n}}|n\rangle \langle n||\psi_{\theta }\rangle$$
$|\phi \rangle$ is a normalized wave function:(22)$$\langle \phi |\phi \rangle =\sqrt{N}\sum_{n=1}^{N}\sqrt{p_{n}}\langle \psi_{\theta }||n\rangle \langle n|\sqrt{N}\sum_{n=1}^{N}\sqrt{p_{n}}|n\rangle \langle n||\psi_{\theta }\rangle =1$$
The density operator $\hat{\rho }$ converges to an average over the outer product of $|\phi \rangle$ under the condition that it is a stochastic wave function.
(23)$$\begin{array}{c} \hat{\rho }=\sum_{n=1}^{N}p_{n}|n\rangle \langle n|=\sum_{n=1}^{N}\sqrt{p_{n}}|n\rangle \langle n|\sum_{n=1}^{N}\sqrt{p_{n}}|n\rangle \langle n|=\sum_{n=1}^{N}\sqrt{p_{n}}|n\rangle \langle n|\;\hat{I}\;\sum_{n=1}^{N}\sqrt{p_{n}}|n\rangle \langle n|=\\ \sum_{n=0}^{N-1}\sqrt{p_{n}}|n\rangle \langle n|\lim_{K\to \infty }\frac{N}{K}\sum_{i=1}^{K}|\psi_{\theta_{i}}\rangle \langle \psi_{\theta_{i}}|\sum_{n=0}^{N-1}\sqrt{p_{n}}|n\rangle \langle n|=\\ \lim_{K\to \infty }\sum_{i=1}^{K}\frac{1}{K}\sqrt{N}\sum_{n=0}^{N-1}\sqrt{p_{n}}|n\rangle \langle n||\psi_{\theta_{i}}\rangle \sqrt{N}\sum_{n=0}^{N-1}\sqrt{p_{n}}\langle \psi_{\theta_{i}}||n\rangle \langle n|=\\ \lim_{K\to \infty }\frac{1}{K}\sum_{i=1}^{K}|\phi_{i}\rangle \langle \phi_{i}|\;\;\;\;\;□ \end{array}$$

As we previously mentioned, a single wavefunction is a pure state and therefore cannot describe a statistical distribution. In particular, a thermal state is never pure. To overcome this issue, a stochastic unraveling method is employed to represent the thermal ancilla qubits that collide with the system in accordance with Equation (Equation 21):(24)$$|\beta_{j}\rangle =\sum_{i=0}^{1}\sqrt{N\frac{e^{-\beta \omega_{i}}}{Z}}|\omega_{i}\rangle \langle \omega_{i}||\psi_{\theta_{j}}\rangle =\sum_{i=0}^{1}\frac{\sqrt{N}e^{-\frac{\beta }{2}\omega_{i}}}{\sqrt{Z}}|\omega_{i}\rangle \langle \omega_{i}||\psi_{\theta_{j}}\rangle$$
where the amplitudes are weighted by square roots of Boltzmann factors. Then, according to Equation (Equation 23), an average of infinite $|\beta_{j}\rangle$ unraveling wavefunctions will converge to a thermal qubit:
(25)$${\hat{\rho }}_{b}=\lim_{K\to \infty }\frac{1}{K}\sum_{j=1}^{K}|\beta_{j}\rangle \langle \beta_{j}|=\frac{e^{-\beta \hat{H}}}{Z}.$$
From here and throughout the paper we will use $|\beta_{j}\rangle$ to describe a stochastic thermal wave function to represent the ancilla.The unraveling of the bath state allows for the unraveling of the collision event. By describing the system as a wavefunction $|\psi_{s}\rangle$, an ensemble average of interactions - described as ${\hat{U}}_{int}$ Equation (Equation 8), between the system and K stochastic thermal wave functions( $|\beta_{j}\rangle$’s) will converge to an interaction between the system in a density matrix representation $|\psi_{s}\rangle \langle \psi_{s}|$ and a thermal qubit state ${\hat{\rho }}_{b}$:(26)$$\begin{array}{c} \lim_{K\to \infty }\frac{1}{K}\sum_{j=1}^{K}{\hat{U}}_{int}|\psi_{s}\rangle ⊗|\beta_{j}\rangle \langle \psi_{s}|⊗\langle \beta_{j}|{\hat{U}}_{int}^{†}=\\ {\hat{U}}_{int}|\psi \rangle \langle \psi |⊗\left(\lim_{K\to \infty }\frac{1}{K}\sum_{j=1}^{n}|\beta_{j}\rangle \langle \beta_{j}|\right){\hat{U}}_{int}^{†}={\hat{U}}_{int}{\hat{\rho }}_{s}⊗{\hat{\rho }}_{b}{\hat{U}}_{int}^{†} \end{array}\;$$
This method enables the description of an interaction with a thermal qubit in the language of wave functions. In order to completely restore Equation (Equation 15), we need to translate the partial trace operation into wave function terminology.

### 2.8. Partial Trace in a Wave Function Description

Partial trace is an essential operation in obtaining the state of a subsystem from a composite state. When the subsystem is entangled with its complementary system, the partial trace operation will lead to a mixed state.

Representing such a state with a single wave function is impossible. Therefore, a stochastic unraveling will be employed. The algorithm is designed to incorporate the correct probabilities in the wave functions such that an average of their outer product will reproduce the reduced subsystem state, i.e., the system after the partial trace operation.

Even though the algorithm presented can be generalized for the tracing out of any number of qubits, in this paper, we present the algorithm of tracing out a single qubit. Specifically, we assume a system of *n* spins and we trace out the $k_{th}$ spin.

For $b\in [0,2^{k-1}-1],\;\;a\in \left[0,2^{n-k}-1\right]$ and i∈[0,1].

We define ${|\phi \rangle }^{i}$
(27)$${|\phi \rangle }_{b\cdot 2^{n-k}+a}^{i}=\frac{{|\psi \rangle }_{\left(2b+i\right)\cdot 2^{n-k}+a}}{N_{i}}$$
The indexes are adopted from a computational basis where *b* and *a* are natural numbers, *n* is the number of particles in the system, *k* is the particle that is traced out, $2\left(b+0\right)\cdot 2^{n-k}+a$ are the indexes of |ψ〉 where the k’th particle is in state 0, and $2\left(b+1\right)\cdot 2^{n-k}+a$ are the indexes of |ψ〉 where the k’th particle is in state 1. N is the normalization factor as well as the square root of the classical probability of this state.

${|\phi \rangle }^{i}$ can be interpreted as a wavefunction reduction caused by the measurement of the environment.

${|\phi \rangle }^{0}$ is a normalized vector of all the elements in $|\psi \rangle$, conditioned on the state of traced-out particle $|0\rangle$.

${|\phi \rangle }^{1}$ is a normalized vector of all the elements in $|\psi \rangle$ conditioned on the state of the traced-out particle $|1\rangle$.

The wavefunction unraveling of the trace becomes:(28)$$tr_{k}\left\{|\psi \rangle \langle \psi |\right\}=\sum_{i=0}^{1}N_{i}^{2}{|\phi \rangle }^{i}{\langle \phi |}^{i}\;\;.$$
The proof is described in Section A.3.

The formulation of Equation (Equation 28) underlines the measurement postulate in quantum mechanics and the equivalence between the partial trace and partial measurement. The partial trace is a sum of a system’s possible states after the traced out particle has “collapsed” to its possible components with the suitable probability.

### 2.9. Stochastic Partial Trace

The main result of Section 2.8 is Equation (Equation 28), developing an unraveling scheme of the mixed state. The dynamical evolution breaks up into separate wavefunctions composing the unraveled mixed state presented in Figure 1. As will be described, the probabilistic nature of the mixed state enables the employment of a Monte Carlo algorithm to randomly choose only one of the unraveling wavefunctions:(29)$$tr_{b_{x_{r}}}|\psi \rangle =\left\{\begin{array}{c} {|\phi \rangle }^{0}\;\mathrm{if}\;x_{r}<N_{0}^{2}\\ {|\phi \rangle }^{1}\;\mathrm{if}\;x_{r}\geq N_{0}^{2} \end{array}\right.\;\;.$$
$N_{0}^{2}$ is the classical probability to find the system in quantum state ${|\phi \rangle }^{0}$ and $N{{}_{1}}^{2}=1-N_{0}^{2}$ is the classical probability to find the system in quantum state ${|\phi \rangle }^{1}$. Thus, for $x_{r}\in \left[0,1\right]$ if $N_{0}^{2}\geq x_{r}$, the Monte Carlo algorithm will induce ${|\phi \rangle }^{0}$. Otherwise, it will induce ${|\phi \rangle }^{1}$. integrating over $x_{r}$ will result in the partial trace $tr_{b}\left\{|\psi \rangle \langle \psi |\right\}$: (30)$$\int_{0}^{1}tr_{b_{x_{r}}}\left\{|\psi \rangle \right\}tr_{b_{x_{r}}}\left\{\langle \psi |\right\}dx_{r}=tr_{b}\left\{|\psi \rangle \langle \psi |\right\}\;\;.$$
Equation (Equation 29) is also a stochastic unraveling, and together with Equation (Equation 26) in Section 2.7, the branching process unraveling is complete.

The branching process described in Figure 1 is a sequence of free dynamics following a unitary collision (see Section 2.5). This process is described in the language of wave functions, allowing an efficient algorithm using wave functions propagation. By unraveling, the algorithm converges to the density operator dynamical representation. By combining the stochastic unraveling and partial trace presented in (Equation 21) and (Equation 27), we can construct Equation (Equation 15), that represent the consecutive collisions of the density matrix with a thermal particle:

Specifically, every wave function $|\psi \rangle$ will undergo three consecutive operations:1.Interaction with a thermal wave function—${\hat{U}}_{int}\left(|\psi \rangle ⊗|\beta \rangle \right)$.2.Stochastic partial trace—$tr_{N+1_{x_{r}}}\left\{{\hat{U}}_{int}|\psi \rangle ⊗|\beta \rangle \right\}$.3.Free dynamic of the system—$\hat{U}tr_{N+1_{x_{r}}}\left\{{\hat{U}}_{int}|\psi \rangle ⊗|\beta \rangle \right\}$.

To accurately restore Equation (Equation 31), $|\psi \rangle$ will accumulate *K* thermal wave functions $|\beta \rangle$(K→∞). The partial trace will yield two different wave functions with different probabilities. Consequently, we will have to average the outer product of all the outcomes with the correct weights. This procedure corresponds to a single collision.
(31)$$\begin{array}{c} \lim_{K\to \infty }\frac{1}{K}\sum_{j=1}^{K}\int_{x_{r}=0}^{1}\hat{U}tr_{{N+1}_{x_{r}}}\left\{{\hat{U}}_{int}|\psi \rangle ⊗|\beta_{j}\rangle \right\}tr_{{N+1}_{x_{r}}}\left\{\langle \psi |⊗\langle \beta_{j}|{\hat{U}}_{int}^{†}\right\}{\hat{U}}^{†}dx_{r}=\\ \;\lim_{K\to \infty }\hat{U}tr_{N+1}\left\{{\hat{U}}_{int}|\psi \rangle \langle \psi |⊗\frac{1}{K}\sum_{j=1}^{K}|\beta_{j}\rangle \langle \beta_{j}|{\hat{U}}_{int}\right\}{\hat{U}}^{†}=\\ \hat{U}\left(tr_{N+1}\left\{{\hat{U}}_{int}\left({\hat{\rho }}_{s}⊗{\hat{\rho }}_{B}\right){\hat{U}}_{int}^{†}\right\}\right){\hat{U}}^{†} \end{array}$$
For repeated collisions, this process will recur for every outcome, as shown in Figure 1. Equation (Equation 15) is restored by consecutively employing Equation (Equation 31) *n* times. To describe this state-to-state map, we define the super-operator:(32)$$\hat{G}\left(\psi ,dt,\phi ,x_{r},\beta \right)=\hat{U}tr_{{N+1}_{x_{r}}}\left\{{\hat{U}}_{int}|\psi \rangle ⊗|\beta_{j}\rangle \right\}$$
A full reconstruction of the process described by Equation (Equation 15) is therefore obtained:
(33)$$\lim_{K\to \infty }\frac{1}{K}\sum_{j=1}^{K}\int_{x_{r}=0}^{1}{\hat{G}}^{n}\left(\psi ,dt,\phi ,x_{r},\beta \right){\hat{G}}^{†n}\left(\psi ,dt,\phi ,x_{r},\beta \right)dx_{r}={\hat{\rho }}_{n}$$

### 2.10. Stochastic Convergence

The direct unraveling process is extremely computationally expensive, because it grows exponentially with each collision as $O(2^{nk})$, where *n* is the number of collisions and *k* is the number of thermal wave functions interacting with each possible system state. This unraveling scheme is described by a branching tree illustrated in Figure 1.

A more efficient computational scheme exploits the stochastic nature of the process in three ways:1.Reduction to a binary tree due to a Wiener process. On the path of a single wavefunction (single branch in the tree in Figure 1), we observe that the system’s wave function interacts at every collision with a stochastic thermal wave function. The process satisfies the conditions of a Wiener process: it has a fixed initial condition and the stochastic part of the bath particle in every collision has a mean 0 and a variance $\sigma^{2}$ [51]. Therefore, for a sufficiently long process, where each collision $U_{int}\left(|\psi_{k}\rangle \langle \psi_{k}|⊗\rho_{b}\right)U_{int}^{†}$ is represented as the average of the outer product of *K* collisions, $U_{int}\left(|\psi_{k}\rangle ⊗|\beta_{j}\rangle \right)$, and thus is split into *K* branches. This process can be sampled by only a single $|\beta_{j}\rangle$. By accumulating many collisions, it will follow the Wiener process, undergo Brownian motion, and the process will converge to a consecutive interaction with a Boltzmann-distributed thermal qubit. As a result, the unraveling of the tree in Figure 2 will converge to the unraveling of the larger tree in Figure 1.As can be seen in Figure 2, this method will generate a properly weighted sample of the binomial distribution around the most probable state.2.The process of tracing out a single bath qubit leads to a mixed state composed of two pure states (see Equation (Equation 28)). Using the property of the Wiener process, the computation of all possibilities with the correct weights will be resolved in a binary tree with changing probability weights. We use a Monte Carlo stochastic partial trace algorithm (Section 2.8) in order to stochastically choose one of the two wave functions composing the mixed state imposed by the partial trace in every step.3.Based on the central limit theorem, the average of a sequence of independent and identically distributed random variables drawn from a distribution of expectation values μ and finite variance $\sigma^{2}$, will converge in probability to a normal distribution. The multidimensional central limit theorem generalizes the theory and states that a random vector (satisfying the vector space axioms) will converge in probability to a multi-variant Gaussian. Mathematically,
(34)$$\sqrt{n}\left(\hat{X_{n}}-\mu \right)\to^{d}N\left(0,\Sigma \right)$$
where Σ is the covariance matrix.

Employing Equation (Equation 33), the average of the outer product of all wave function possibilities with the adequate probabilities represented by the tree in Figure 1 converges to the density matrix $\rho_{n}$, satisfying the collision model in Section 2.5. Thus, a sample of the outer products of *K* identical wave functions undergoing n consecutive collisions by Equation (Equation 33): $G^{n}\left(\psi ,dt,\phi ,x_{r},\beta \right){G^{n}}^{†}\left(\psi ,dt,\phi ,x_{r},\beta \right)$ is a sequence of independent and random variables drawn from a distribution of expected value $\rho_{n}$ and a finite variance. Thus,
(35)$$\lim_{K\to \infty }\frac{1}{K}\sum_{i=1}^{K}G^{n}\left(\psi ,dt,\phi ,x_{r},\beta \right){G^{n}}^{†}\left(\psi ,dt,\phi ,x_{r},\beta \right)={\hat{\rho }}_{n}\;\;.$$
Moreover, with the number *n* of collisions increasing, we expect a convergence in probability if K is finite.
(36)$$\frac{1}{K}\sum_{i=1}^{K}G^{n}\left(\psi ,dt,\phi ,x_{r},\beta \right){G^{n}}^{†}\left(\psi ,dt,\phi ,x_{r},\beta \right)-{\hat{\rho }}_{n}\to^{d}N\left(0,\frac{\Sigma }{K}\right)$$
The central limit theorem is redundant with the mathematical description of Brownian motion. Yet, we have found it useful for physical insight into the process.

## 3. Results

### 3.1. Convergence

To illustrate the unraveling approach in accordance with Section 2.10, a specific example of a unitary interaction is explored. An interaction leading to a partial swap between the last particle in a system of qubits and an uncorrelated thermal qubit are specifically chosen. This type of interaction has been addressed in the collision model reviewed by Ciccarello, Lorenzo, Giovannetti, and Palma [52].

In a two-qubit system, the swap algorithm becomes the swap gate:(37)$$\hat{S}=|0\rangle \langle 0|⊗|0\rangle \langle 0|+|1\rangle \langle 1|⊗|1\rangle \langle 1|+|1\rangle \langle 0|⊗|0\rangle \langle 1|+|0\rangle \langle 1|⊗|1\rangle \langle 0|=\left(\begin{array}{cccc} 1 & 0 & 0 & 0\\ 0 & 0 & 1 & 0\\ 0 & 1 & 0 & 0\\ 0 & 0 & 0 & 1 \end{array}\right)$$
In a larger system composed of *N* qubits, an operation that swaps between the ith and jth qubits, the swap algorithm is similar:

Let
(38)$$\prod_{k=0}^{i-1}⊗I_{2^{k}}⊗|\delta \rangle ⊗\prod_{k=i+1}^{N}⊗I_{2^{k}}=|\delta_{i}\rangle$$
(39)$$\prod_{k=0}^{i-1}⊗I_{2^{k}}⊗\langle \delta |⊗\prod_{k=i+1}^{N}⊗I_{2^{k}}=\langle \delta_{i}|.$$
A swap operation between particles i and j can be described as:(40)$${\hat{S}}_{i,j}=|0_{i}\rangle \langle 0_{i}||0_{j}\rangle \langle 0_{j}|+|1_{i}\rangle \langle 1_{i}||1_{j}\rangle \langle 1_{j}|+|0_{i}\rangle \langle 1_{i}||1_{j}\rangle \langle 0_{j}|+|1_{i}\rangle \langle 0_{i}||0_{j}\rangle \langle 1_{j}|.$$
The Swap is a unitary operation—see Section A.1:(41)$$\hat{S}{\hat{S}}^{†}={\hat{S}}_{k,m}{\hat{S}}_{k,m}^{†}={\hat{I}}_{2^{n}}$$
$\hat{S}$ is unitary and real, therefore, $\hat{S}={\hat{S}}^{†}$.

The unitarity of the swap operator renders it a preferable candidate for simulating interactions. Its advantage lies in its ability to be realized on a quantum computer.

A partial swap is defined as:(42)$${\hat{S}}_{p}=\cos\left(\theta \right){\hat{I}}_{2^{n}}+i\sin\left(\theta \right)\hat{S}\;\;,$$
where θ defines the degree of mixing. The partial swap is also a unitary operation ${\hat{S}}_{p}{\hat{S}}_{p}^{†}=\hat{I}$ (see Section A.2). Unlike the full swap, the partial swap induces a spectrum of interaction strengths and maintains a correlation between all particles in the system after a reduced description of the setup, as we have seen in Section 2.4. For this reason, we have chosen it to be a preferable candidate for the interaction between the ancilla and the system.

Because $S_{p}$ is unitary, by definition, we can write:(43)$${\hat{S}}_{p}=e^{-i{\hat{H}}_{int}\theta }$$
θ can be either a phase or angle of interaction with units of time. Because we can write an exponent as a polynomial sum of powers of ${\hat{H}}_{int}$, we obtain
(44)$$[{\hat{S}}_{p},{\hat{H}}_{int}]=0\;\;\;.$$

We now define the map generated by the unraveling of K processes acting on a state |ψ〉 using the super operator G defined in Equation (Equation 32): (45)$$\frac{1}{K}\sum^{K}G^{n}\left(|\psi \rangle \right){G^{n}}^{†}\left(\langle \psi |\right)=\Theta_{K_{n}}\left(|\psi \rangle \langle \psi |\right)\;$$


When $\rho_{0}=|\psi_{0}\rangle \langle \psi_{0}|$ and $\rho_{n}$ is the density matrix $\rho_{0}$ undergoing dynamics according to the map in Equation (Equation 15). We expect the distance
(46)$$D=\left|\right|\Theta_{K_{n}}\left(\rho_{0}\right)-\rho_{n}\left|\right|$$
to converge in probability to $N(0,\frac{\Sigma }{K})$: a normal distribution around zero. In order to examine the convergence rate of our model, we choose to observe the variance of the distance distribution and expect it to converge to $\frac{1}{K}$. The chosen distance function is the variance between each element of the unraveling and the density matrix $\rho_{s}$ of the system undergoing the same dynamics in the density matrix form of Equation (Equation 15) and in the wavefunction unraveling Equation (Equation 36). $N^{2}$ is the dimension of the system, *K* is the number of realizations, and *n* the number of collisions.
(47)$$D\left(G,\rho ,n,N,K\right)=\frac{1}{N^{2}}\sum_{i,j=1}^{N^{2}}{\left|\rho_{n_{i,j}}-\Theta_{K_{n_{i,j}}}\left(\rho_{0}\right)\right|}^{2}$$

According to the multidimensional central limit theorem, we expect a convergence of the covariance matrix to $\frac{1}{K}$, and thus the variance of every element $|\Theta_{i,j}-\rho_{i,j}|$ to $\sigma_{i,j}^{2}$. Because each element of Θ has the same dependency on the stochastic variable, we expect $\sigma_{i,j}=\sigma_{k,l}$. Thus, we expect *D* to approximate the average of all element variances.
(48)$$D\left(G,\rho ,n,N,K\right)=\frac{1}{N^{2}}\sum_{i,j=1}^{N}|\rho_{n_{i,j}}-\Theta_{K_{n_{i,j}}}\left(\rho_{0}\right){|}^{2}\approx \frac{1}{N^{2}}\sum_{i,j}^{N}\frac{1}{K}\sum_{k}^{K}{\left|\rho_{n_{i,j}}-\Theta_{K_{n_{i,j_{k}}}}\left(\rho_{0}\right)\right|}^{2}.$$
From the central limit theorem, we expect that the sum of all element variances will decay as $\frac{1}{K}$ with *K* as the number of realizations:(49)$$\sum_{i,j}^{N^{2}}\frac{1}{K}\sum_{k}^{K}{\left|\rho_{n_{i,j_{k}}}-\Theta_{K_{n_{i,j_{k}}}}\left(\rho_{0}\right)\right|}^{2}∼\frac{1}{K}\;\;.$$
Thus
(50)$$D\left(G,\rho ,n,N,K\right)∼\frac{1}{K}$$
As we can see in Figure 3, we observe that *D* indeed converges as expected.

## 4. Conclusions

In this paper, we developed the basic tools for unraveling open system dynamics with wavefunctions based on a collision model. The stochastic averaging of these wavefunctions converges for all expectation values to those obtained by the density operator formalism. The basic algorithm can be divided into three steps:1.Implementing a Monte Carlo stochastic partial trace algorithm (Section 2.8 and Section 2.9).2.Restoring a mixed state by unraveling of stochastic wave functions. This was achieved for the bath particle colliding with the system and for the system itself after averaging many stochastic system wave functions undergoing dynamics (Section 2.7).3.Using statistical properties, such as the central limit theorem, we observed convergence to the dynamics represented by a density matrix. The convergence was achieved with a relatively small number of realizations. This property resulted in high computational efficiency.

An illustration of the algorithm was provided in Section 3.1.

The algorithm developed contains a non-linear component. In the implementation of the Monte Carlo algorithm in Section 2.9, one of the wave functions composing the mixed state was selected by a partial trace. The probability of choosing each of the states was calculated from the normalization factor of one of the states. This required first to calculate one possible outcome, which might not be used in the following step. If the selected wavefunction was not the one that was calculated, the other wavefunction had to be recalculated. Thus, the non-linearity resulted from renormalizing the wavefunction and the possibility of computing an additional wavefunctions when it was absent in the Monte Carlo lottery.

The modeling method addresses a major problem of the computation cost of simulating open quantum systems. The wavefunction representation reduces the memory requirement, and as the system becomes larger, the speed of convergence to the full simulation also increases. We therefore expect a reduction in computational cost of up to a factor of *N*, where *N* is the size of Hilbert space. As a result, the possible simulations boundaries are stretched.

In addition, the wavefunction method allows insight into the ongoing physical process from a single-event or quantum trajectory viewpoint. [53]

Finally, the simulation can be implemented on existing quantum computers. These, in turn, possess unitary operations performed on wavefunctions.

## Figures and Tables

**Figure 1 entropy-24-01808-f001:**
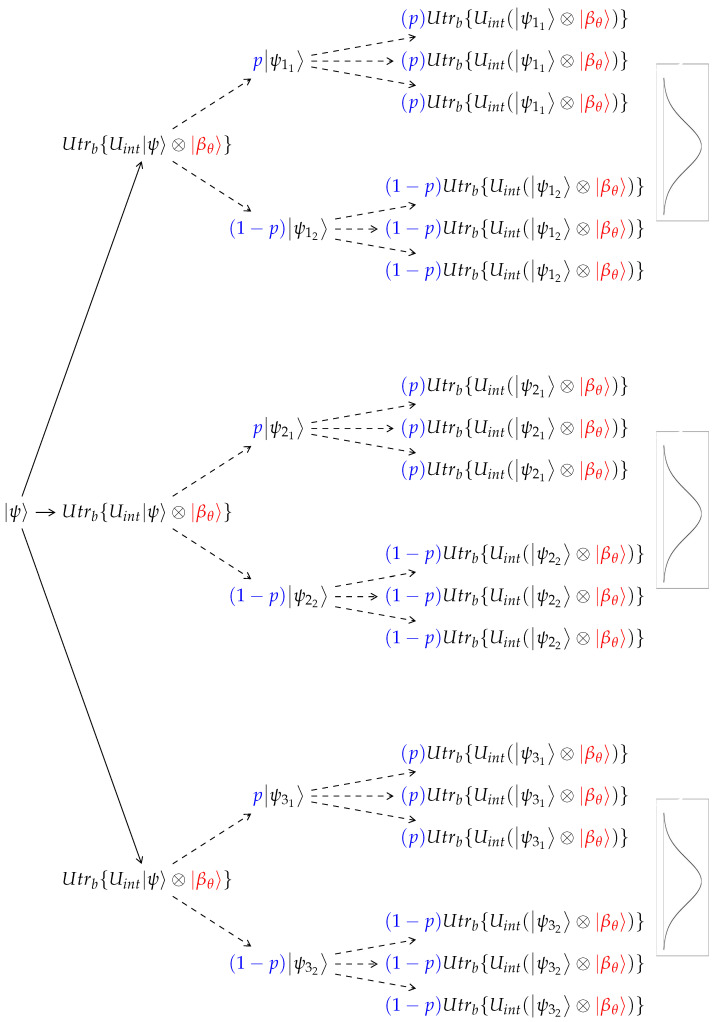
The unraveling tree. The time evolution of the system’s wave function $|\psi \rangle$. Three different interactions with an ancilla $|\beta_{\theta }\rangle$ are presented. Each interaction spawns two wave functions with different weights. This process is repeated with each ancilla interaction. The Gaussian distribution demonstrates that the process asymptotically obeys the central limit theorem.

**Figure 2 entropy-24-01808-f002:**
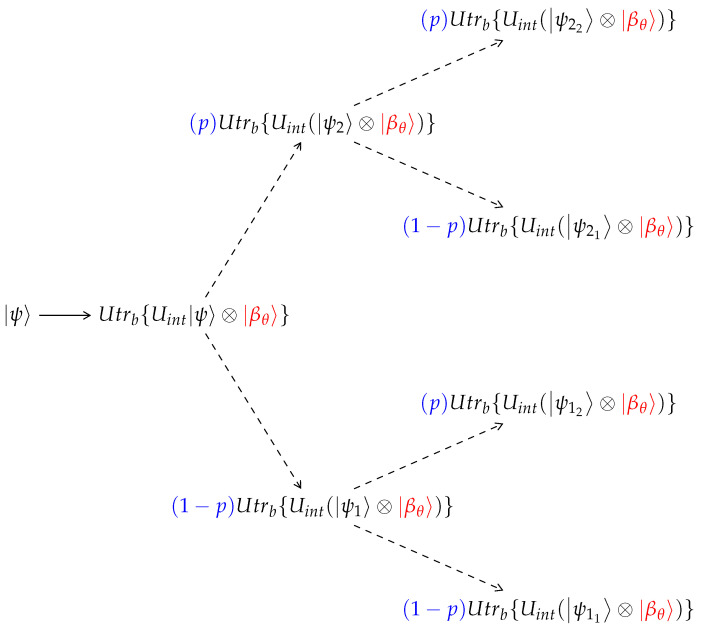
The reduced unraveling tree. Interaction with a thermal qubit is in agreement with the definitions of Wiener process. Thus, the tree can be reduced to a binary tree form.

**Figure 3 entropy-24-01808-f003:**
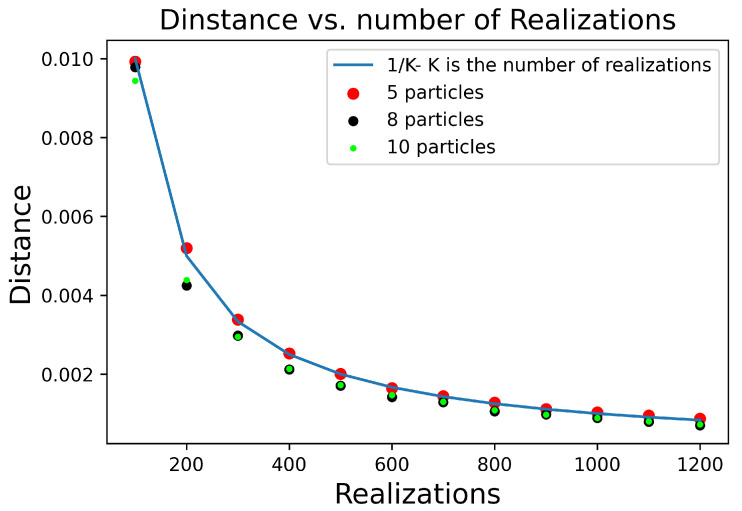
The distance function D is the normalized sum of the absolute value squared of the Euclidean distance between the same elements of $\rho_{n}$, and $\Theta_{K_{n}}\left(\rho_{0}\right)$. D is defined as $D\left(G,\rho ,n,N\right)=\frac{1}{N^{2}}\sum_{i=1}^{N^{2}}{\left|\rho_{n_{i,j}}-\Theta_{K_{n_{i,j}}}\left(\rho_{0}\right)\right|}^{2}$. We expect that D will converge as $\frac{1}{K}$. Because the variance should converge as $\frac{1}{K}$, we expect the variance of each element also to converge as $\frac{1}{K}$. Therefore, D, that is, the sum of variance element of each matrix element, will also converge as $\frac{1}{K}$. Figure 3 exhibits the D function for three systems undergoing 600 collisions. We can observe a decay to zero as $\frac{1}{K}$ for a system composed of 5, 8, and 10 particles, as expected.

## Data Availability

Not applicable.

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
