# Peer review of "Wave Function Realization of a Thermal Collision Model"

_entropy, 2022, doi:10.3390/e24121808_

Round 1

Reviewer 1 Report

In the abstract the authors claim to present an efficient algorithm to simulate open system dynamics. This algorithm is based on quantum collision model and stochastic unraveling of the wave function.

The introduction of the paper is actually quite confusing. Although several concepts are recalled, some of them even repeatedly, not all of them seemingly related to the presented work. Moreover, some of these concepts are very badly stated, and some of them are presented without necessary citations or explanations.

Furthermore, the quality of the English is far from excellent and many phrases are logically disconnected, which further hinders the understanding of the work.

As for the technical part of the work, notation is kind of confusing, as for instance it makes difficult to understand whether a certain wave function is describing a spin in the open quantum system or an ancilla. Moreover most equations are not explained, neither from a notational or a physical point of view, making it hard to understand.

It is also not clear in which sense a collision model is used, as the only dynamical equation provided for the density matrix is Eq.(12) which is a time-continuous equation, while a collision model is ineherently time-discrete, unless a continuous time limit is taken, which is not the case in the present work.

To conclude, in its present state the work is hard to understand due to linguistic, mathematical and notational problems. As long as statements are not made clearer and the notation and mathematical equations well explained, it is hard to even judge the correctness or significance of the results.

I thus invite the authors to deeply revise this work before even considering for publication.

Reviewer 2 Report

In the paper, the authors have presented an elegant methodology for simulating open quantum systems, combining the simplicity of a collisional model and the efficiency of a stochastic wave function unraveling. The authors also illustrated the implementation and analyzed the convergence of the approach in details. I strongly believe that this paper will contribute to the development of collisional models which have already become versatile tools in the field of quantum thermodynamics. Hence I recommend the publication of the paper in its present form.

Round 2

Reviewer 1 Report

In this revised version of the manuscript, the authors successfully addressed the main issues raised in the previous version. In the present form the manuscript is clearly understandable, and I can now assert that the algorithm presented is effectively an interesting proposal.

Only minor typos are present, which anyway do not represent an obstacle to the reading.

Few minor remarks have still to be made:

-I suggest adding the following relevant references:

The following two on the connection between collision models and quantum trajectories

Brun, T.A. A simple model of quantum trajectories. Am. J. Phys. 2002, 70, 719–737

Gross, J.A.; Caves, C.M.; Milburn, G.J.; Combes, J. Qubit models of weak continuous measurements: Markovian conditional and open-system dynamics. Quantum Sci. Technol. 2018, 3, 024005.

The following two on the use of collision models in thermodynamics:

Cusumano, S.; Cavina, V.; Keck, M.; Pasquale, A.D.; Giovannetti, V. Entropy production and asymptotic factorization via thermalization: A collisional model approach. Phys. Rev. A 2018, 98, 032119

Strasberg, P.; Schaller, G.; Brandes, T.; Esposito, M. Quantum and Information Thermodynamics: A Unifying Framework Based on Repeated Interactions. Phys. Rev. X 2017, 7, 021003.

The following two on multipartite systems:

Cusumano, S.; Mari, A.; Giovannetti, V. Interferometric quantum cascade systems. Phys. Rev. A 2017, 95, 053838.

Cusumano, S.; Mari, A.; Giovannetti, V. Interferometric modulation of quantum cascade interactions. Phys. Rev. A 2018, 97, 053811.

The following one in connection of Monte Carlo methods in open quantum systems:

Dum, R.; Zoller, P.; Ritsch, H. Monte Carlo simulation of the atomic master equation for spontaneous emission. Phys. Rev. A 1992, 45, 4879–4887.

-I find that Figures 1,2 would better serve their scope if put later in the manuscript, specifically in Sec. 2.9 or 2.10

-Though I agree it is true, it would be good to cite some reference where the lemma in Eq.17 is demonstrated (even a standard textbook is fine).

-Does Eq.23 refer to an ancilla state? If yes, better substitute \rho with \rho_b. If not, then it is better to explain to what that state refers.

--The tracing procedure in Eq.27 should be explained a little more in depth, specifically spending some time on the definition if the symbols and explaining in detail the procedure. 

-The distance in Eq.46 is not properly defined. It should be explained better how it is defined. Also, I probably misunderstood, but it looks like \Theta_n(K) is a map, while \rho_n is a state, so it is not clear how to make the difference between the two.

Once these minor remarks have been solved, I believe the manuscript will be good for being accepted in Entropy.
